# Magnetic-Responsive Doxorubicin-Containing Materials Based on Fe_3_O_4_ Nanoparticles with a SiO_2_/PEG Shell and Study of Their Effects on Cancer Cell Lines

**DOI:** 10.3390/ijms23169093

**Published:** 2022-08-13

**Authors:** Alexander M. Demin, Alexander V. Vakhrushev, Alexandra G. Pershina, Marina S. Valova, Lina V. Efimova, Alexandra A. Syomchina, Mikhail A. Uimin, Artem S. Minin, Galina L. Levit, Victor P. Krasnov, Valery N. Charushin

**Affiliations:** 1Postovsky Institute of Organic Synthesis, Russian Academy of Sciences (Ural Branch), 620108 Ekaterinburg, Russia; 2Center of Bioscience and Bioengineering, Siberian State Medical University, 634050 Tomsk, Russia; 3Research School of Chemistry & Applied Biomedical Sciences, National Research Tomsk Polytechnic University, 634050 Tomsk, Russia; 4Biological Institute, National Research Tomsk State University, 634050 Tomsk, Russia; 5Mikheev Institute of Metal Physics, Russian Academy of Sciences (Ural Branch), 620990 Ekaterinburg, Russia; 6Institute of Chemical Engineering, Ural Federal University, 620002 Ekaterinburg, Russia

**Keywords:** magnetic nanoparticles, SiO_2_, PEG, doxorubicin, alternating magnetic field, tumor cells, drug-delivery

## Abstract

Novel nanocomposite materials based on Fe_3_O_4_ magnetic nanoparticles (MNPs) coated with silica and covalently modified by [(3-triethoxysilyl)propyl]succinic acid–polyethylene glycol (PEG 3000) conjugate, which provides a high level of doxorubicin (Dox) loading, were obtained. The efficiency of Dox desorption from the surface of nanomaterials under the action of an alternating magnetic field (AMF) in acidic and neutral media was evaluated. Their high cytotoxicity against tumor cells, as well as the drug release upon application of AMF, which leads to an increase in the cytotoxic effect, was demonstrated.

## 1. Introduction

One of the priority tasks of modern nanomedicine is the design of new, highly efficient delivery vehicles for anticancer agents. For this purpose, mesoporous SiO_2_-based nanomaterials with high biocompatibility and sorption capacity are often exploited [1,2]. The use of magnetic nanoparticles (MNPs) in such materials makes it possible to expand the range of their properties; in particular, it allows visualization of their biodistribution and accumulation in tissues by magnetic resonance imaging (MRI) [3,4,5,6,7,8] or magnetic particle imaging (MPI) [9], as well as the enhancement of their therapeutic action due to targeting and concentrating of magnetic nanocomposites in cancer cells by external magnetic stimulus [10] or due to the hyperthermia effect caused by the heating of magnetic core when an alternating magnetic field (AMF) is applied [11]. At the same time, the therapeutic effect can be achieved not only by heating the tissues in which MNPs are localized but also by the increased release rate of antitumor agents due to local heating of the cores of nanocomposite materials or through a synergistic effect when hyperthermia and chemotherapy are combined [12,13,14,15]. MNPs are also used in the design of hybrid antitumor drug delivery systems to enable controlled release using high intensity-focused ultrasound (HIFU) technique [7].

An important factor in the design of nanomaterials for medical applications is the choice of an additional coating. Thus, at present, in addition to the mesoporous SiO_2_ shell, coatings based on chitosan [16,17], polyvinyl alcohol (PVA) [17], k-carrageenan [18], polyamidoamine (PAMAM) dendrimers [19], polyethyleneimine (PEI) [20,21], tannic acid [22], etc. are used. One of the biocompatible polymers, which is the most commonly used for this purpose, is polyethylene glycol (PEG) [6,17,23], the molecules of which are known to increase the aggregative stability of nanoparticles and impart a “stealth” effect to the particles, thus preventing their recognition by mononuclear phagocytes [24]. As a result, it becomes possible to increase the time of nanomaterial circulation in the blood and enhance the therapeutic effect.

Doxorubicin (Dox) is the most commonly prescribed anticancer agent. It is known that its molecule has a positive charge (p*K*_a_ = 8.2) in aqueous media with neutral pH. Therefore, the design of materials with a negatively charged surface in order to increase the efficiency of Dox sorption appears to be the most promising approach.

The purpose of this study was to develop new magnetic-responsive SiO_2_/PEG-coated nanomaterials with a high negative surface charge as potential Dox delivery vehicles to evaluate the efficiency of Dox desorption under the action of AMF in acidic and neutral media and to access their cytotoxicity against cancer cells in vitro.

## 2. Results and Discussion

### 2.1. Synthesis and Characterization of Nanocomposite Materials

The initial MNPs were obtained by co-precipitation from a solution of Fe^3+^ and Fe^2+^ salts, as described previously [25]. The application of SiO_2_ coating was carried out by the sol-gel method using a 1.5 molar excess of tetraethoxyorthosilicate (TEOS); as a result, MNPs **1** were obtained [3,26]. Alkoxysilane derivatives containing, for example, amino, glycidoxy, mercapto, isocyanate, vinyl, or other groups are often used for covalent attachment of biologically active or functional molecules to the SiO_2_ surface of nanoparticles [18,27,28,29,30,31]. In this work, we have exploited the strategy of covalent binding of a modified PEG molecule with terminal amino and carboxyl groups to the surface of MNPs **1** using [(3-triethoxysilyl)propyl]succinic anhydride (TESPSA) [32] as an auxiliary reagent (Figure 1). This reagent has three ethoxysilyl groups, which, after hydrolysis, are able to form a covalent bond with the SiO_2_-modified surface of MNPs **1**, and the anhydride moiety capable of coupling to the amino group of *O*-(2-aminoethyl)-*O*’-(2-carboxyethyl)polyethylene glycol 3000 to form the amide bond (Figure 2).

To find the optimum reaction conditions, we performed preliminary experiments on the surface modification of MNPs **1** using different amounts of TESPSA: 3.0, 0.3, 0.03 mmol per 1 g of MNP@SiO_2_ (Figure 1) and estimated the presence of succinic acid propyl silyl (SAPS) residues on the MNPs surface by IR spectroscopy, energy-dispersive X-ray (EDX) spectroscopy and elemental analysis (EA) data (Figure 1a, Appendix A).

The presence of SAPS residues could be reliably identified only in the MNPs modified by the largest TESPSA amount (3.0 mmol per 1 g of MNPs) due to the bands in the region of 1708 and 1660 cm^−1^, which correspond to the stretching vibrations of free and protonated C = O groups, respectively.

Anhydrides are active compounds and can undergo hydrolysis or esterification under silanization conditions (in 70% EtOH). Therefore, we have optimized the amidation reaction of the PEG derivative with TESPSA. For this, the PEG derivative was added to a solution of TESPSA in dry DMSO-d_6_ in a 1:1 molar ratio. The course of the amidation reaction was monitored using ^1^H NMR spectroscopy by an increase in the intensity of the signal at *δ* 7.8 ppm corresponding to the amide proton of the formed bond (similarly to Ref. [32]). It was found that the formation of conjugate **5** (Figure 2) was completed within 2 h. The hydrolysis of ethoxysilyl groups proved to occur simultaneously due to the presence of traces of water in DMSO-d_6_, which, with an increase in the exposure time, would lead to complete hydrolysis and the polycondensation processes. Accordingly, the efficiency of silane binding to the surface of MNPs **1** has to decrease. Therefore, when studying the coupling reaction by ^1^H NMR spectroscopy, we also estimated the rate of hydrolysis of the TESPSA ethoxysilyl groups by the decrease in the intensity of the proton signals from CH_3_CH_2_ (*δ* 3.74 ppm), CH_3_CH_2_ (*δ* 1.40 ppm), and ≡Si–CH_2_ groups (*δ* 0.56 ppm) (Appendix A) (similarly to Ref. [32]). Thus, after 2 h, only 44% of the ethoxy groups of the starting TESPSA remained to be registered in the reaction mixture. Thus, to modify nanoparticles, solutions of TESPSA and PEG were mixed in dry DMSO, and after 1 h, without isolating conjugate **5**, they were added to a suspension of MNPs in 70% EtOH and stirred for another 20 h.

To optimize the PEGylation of MNPs and obtain nanoparticles with optimal hydrodynamic and sorption properties, we performed synthesis varying MNPs **1**/conjugate **5** weight ratios (Figure 2) and obtained MNPs **6**–**8**, respectively. The fixation of conjugate **5** on the surface of MNPs **1** proved to occur due to the formation of a Si–O–Si bond. Immobilization of conjugate **5** was confirmed by the ATR-FTIR spectroscopy (Figure 2a), EDX spectroscopy, as well as EA data (Appendix A).

In the IR spectra of MNPs **6**–**11** (Figure 2a), the absorption bands in the region of 1660 and 1440 cm^−1^, characteristic of stretching and bending vibrations of the PEG C = O and C–H groups, respectively, were observed. As follows from the characteristic band intensities and elemental analysis data (Appendix A), MNPs **6** contain the largest amount of conjugate **5**. According to DLS data (Figure 2b), all synthesized nanoparticles acquire better hydrodynamic characteristics (the average hydrodynamic diameter *D*_h_, polydispersity index PdI and ζ-potential ZP) compared to the initial particles. It is worth noting that in addition to the PEG carboxyl group, another carboxyl group was formed on the particle surface due to the coupling of the succinic anhydride fragment to the PEG amino group (Figure 2). This resulted in particles with a higher negative charge (Figure 2b), which may contribute to an increase in Dox loading on the surface of these materials.

Sorption of Dox onto the surface of MNPs **6**–**8** was carried out in a way described in [33]. For this, a Dox solution was added to an aqueous colloidal solution of MNPs **6**–**8** and stirred for 20 h; then, the resulting MNPs **9**–**11** were precipitated by centrifugation (Figure 2). In the IR spectra of MNPs **9**–**11** (Figure 2a), there were well-defined absorption bands characteristic of Dox: 1725 cm^−1^ (ν С = О of the side chain), 1640 cm^−1^ (ν С = О in the anthracene fragment and PEG COOH), 1585 cm^−1^ (δ N–H), group of bands at 1447, 1416, and 1381 cm^−1^ (δ C–H).

The efficiency of Dox sorption on MNPs **9**–**11**, namely, Dox loading efficiency (LE) and loading capacity (LC), was estimated using UV spectrometry by a decrease in the Dox concentration in supernatants after the particles were removed by centrifugation from the reaction mixtures. It has been shown that Dox LC is 16.2, 14.6, and 13.3% for MNPs **9–11**, respectively, thus exceeding the loading level of a number of mesoporous SiO_2_ materials (for example, LC for mesoporous SiO_2_ nanoparticles was 8.4% [21] or 2.9% [34]) and MNP@SiO_2_ materials (for example, LC for MNPs Fe_3_O_4_@SiO_2_@Tannic acid loaded simultaneously with methotrexate and Dox was 7.6 and 3.2%, respectively [22]).

The efficiency of Dox sorption on MNPs **9**–**11**, namely, Dox loading efficiency (LE) and loading capacity (LC), was estimated using UV spectrometry by a decrease in the Dox concentration in supernatants after the particles were removed by centrifugation from the reaction mixtures. It has been shown that Dox LC is 16.2, 14.6, and 13.3% for MNPs **9–11**, respectively, thus exceeding the loading level of a number of mesoporous SiO_2_ materials (for example, LC for mesoporous SiO_2_ nanoparticles was 8.4% [21] or 2.9% [34]) and MNP@SiO_2_ materials (for example, LC for MNPs Fe_3_O_4_@SiO_2_@Tannic acid loaded simultaneously with methotrexate and Dox was 7.6 and 3.2%, respectively [22]).

Thus, MNPs **9** turned out to be optimal both in terms of loading level and hydrodynamic characteristics (Figure 2b), and, accordingly, they appear to be promising for further research. As an example, Figure 2c shows TEM images of MNPs **6** and **9**. As can be seen, MNPs do not undergo significant changes in shape and size as a result of Dox immobilization. MNPs **9** had an average diameter of 13 nm and a coating thickness of 1.5–2.5 nm. The particle size distribution of MNPs **6** and **9** is presented in Appendix A. The electron diffraction data confirmed the presence of the magnetite structure (JCPDS Card No. (79-0417)) in the obtained MNPs (Figure 2c (inset), Appendix A and Appendix A). To study the cytotoxic effect, including that caused by the action of an external high-frequency AMF, we chose MNPs **9**.

The synthesized MNPs **6** and **9** had high values of specific magnetization (*M*_s_) (Figure 3a), which indicates a high magnetic response to the magnetic field.

It is known that magnetic particles are capable of generating heat under AMF application; this phenomenon is called magnetic hyperthermia [11,35]. The specific absorption rate (SAR) (or specific loss power (SLP)) and intrinsic loss power (ILP) are the main parameters for characterizing the heat dissipation efficiency of MNPs in AMF. MNPs **6** were shown to heat up to 45 °C in 10 min when exposed to a magnetic field of maximum strength *H* (0.27 kOe) and frequency *f* (230 kHz) (Figure 3b). The values of SAR and ILP of MNPs **6** were calculated for their colloidal solution at a concentration of 10 mg[Fe]/mL (Figure 3b).

### 2.2. MTT Cytotoxicity Assay

We examined the cytotoxic effect (MTT assay) on various human (Appendix A) and mouse (Appendix A) tumor cell lines (MDA-MB231, HepG2, 4T1, CT26, and B16) and compared the IC_50_ of Dox and MNPs **6** and **9** under study (Table 1 and Table 2). To do this, we incubated the studied materials with cells for 24 and 48 h. It has been shown that the sensitivity of various cell lines to the cytotoxic effect of MNPs corresponds to their sensitivity to free Dox and the calculated IC_50_ values are comparable (Table 2). Thus, we can conclude that cell death is attributed to the release of Dox from the MNPs surface. For the parent MNPs **6**, the IC_50_ was more than 10 times higher (Table 1), so they can be classified as nontoxic.

### 2.3. Dox Release under Exposure to AMF

We studied the release of Dox in media with pH = 5.8 and 7.4, depending on the modes of AMF application: short pulses for 1 h (1 min exposure/1 min pause); 10 min pulses for 2 h (10 min exposure/10 min pause); continuous exposure to AMF for 1 h. The amount of released Dox was determined by fluorescence spectrometry (λ_ex_ = 480 nm, λ_em_ = 590 nm). The latter AMF mode proved to be the most efficient, enhancing desorption of Dox at pH = 5.8. The level of desorption was comparable to the amount of Dox released under continuous stirring for 24 h (Figure 3c). Based on this, it can be concluded that conditions (continuous exposure to AMF for 1 h), which ensure the constancy of the particle temperature at the maximum level, make it possible to achieve the best results of Dox desorption.

Then, we studied the effect of AMF on the cytotoxicity of MNPs **9** against 4T1 cells. Notably, in the absence of AMF, the cytotoxicity of MNPs **6** and **9** was not statistically different (Figure 3d). However, it has been demonstrated that cell death in the presence of MNPs **9** is enhanced by a factor of 2.5 when applying AMF (*H =* 0.27 kOe, *f =* 230 kHz) close to the safety limit (*H* × *f*) of 6.25 × 10^7^ Oe Hz [35] (Figure 3d). We did not observe heating of colloidal solutions of MNPs **1** at a concentration of less than 1 mg [Fe]/mL above 37 °C, so we can assume that in the case of in vitro experiments with MNPs **9** at a concentration of 5 µg [Fe]/mL, the cytotoxicity is due to the increased Dox desorption rather than the hyperthermia of the sample as a whole. Thus, the AMF application leads to local heating of the particle core only, which accelerates the Dox release with a subsequent increase in cellular toxicity (Figure 3d). The same “local magnetic hyperthermia” or “hot spot” effect was demonstrated earlier, for example, in [36,37,38,39]. Such conditions are more preferable for further medical applications since local heating of MNPs does not lead to undesirable effects caused by overheating, such as tissue necrosis and inflammatory reactions in tissues. Moreover, the pronounced AFM-stimulated cytotoxic effect is realized at a relatively low MNPs concentration that, for example, is attainable in the tumor after intravenous injection.

## 3. Materials and Methods

### 3.1. Materials

We used FeCl_3_ × 6H_2_O and FeSO_4_ × 7H_2_O (Sigma-Aldrich, St. Louis, MO, USA), tetraethoxysilane (TEOS, Alfa Aesar, Heysham, Lancashire, UK), [(3-triethoxysilyl)propyl]succinic anhydride (TESPSA, TCI Europe, Zwijndrecht, Belgium), *O*-(2-aminoethyl)-*O*’-(2-carboxyethyl)polyethylene glycol 3000 (PEG, Sigma-Aldrich, St. Louis, MO, USA) and doxorubicin hydrochloride (Dox, Sigma-Aldrich, St. Louis, MO, USA).

### 3.2. Synthesis of MNPs with SiO_2_ Shell (MNPs ***1***)

A saturated solution of NH_4_OH (4.5 mL) was added to 45 mL of an aqueous solution of FeCl_3_ × 6H_2_O (1.051 g, 3.89 mmol) and FeSO_4_ × 7H_2_O (0.540 g, 1.94 mmol) under sonification at 40 °С (as described in Refs. [25,26]). After 10 min, nanoparticles were precipitated with a magnet, washed with water (3 × 30 mL), and suspended in water (40 mL) to afford a suspension of Fe_3_O_4_ MNPs.

EtOH (75 mL) was added to a suspension of Fe_3_O_4_ MNPs (0.182 g, 0.787 mmol) in water (15 mL); the reaction mixture was heated to 40 °С, then a solution of TEOS (0.265 µL, 1.18 mmol) in EtOH (20 mL) was added under ultrasound stirring, and a saturated solution of NH_4_OH (2.6 mL) was added dropwise for 10 min (by analogy with Ref. [26]). Stirring was continued for 20 h at 25 °С. The resulting nanoparticles were separated by centrifugation (40,000 g, 10 min), washed with water (3 × 60 mL), and dispersed in water (18 mL) to afford suspensions of MNPs **1**.

### 3.3. Synthesis of MNPs@TESPSA (MNPs ***2**–**4***)

EtOH (5 mL) was added to a colloidal solution of MNPs **1** (20 mg) in water (2 mL); then, a solution of TESPSA (60, 6, or 0.6 µmol) in DMSO (3 mL) was added to the resulting suspensions at 40 °C. The reaction mixture was sonicated in an ultrasonic bath and stirred with an overhead stirrer for 4 h at 40 °C and 16 h at 20 °C. The resulting nanoparticles were separated by centrifugation (10,000× g, 10 min), washed with water (3 × 10 mL), and dispersed in water (5 mL) to afford suspensions of MNPs@TESPSA **2**–**4**, correspondingly.

### 3.4. Synthesis of Conjugate ***5*** NMR Experiment

TESPSA (0.51 mg, 1.6 µmol) was added to a solution of PEG (5 mg, 1.6 µmol) in DMSO-d_6_ (1.1 mL). The ^1^H NMR spectra were recorded at ambient temperature in 2, 6, and 25 h.

**TESPSA**. ^1^H NMR (DMSO-d_6_, 400 MHz) *δ* 3.74 (6H, q, *J* = 7.0 Hz, 3CH_2_ (OEt)), 3.26–3.17 (1H, m, CH), 3.04 (1H, dd, *J* = 18.1, 9.7 Hz, CH_a_ (succinic anhydride CH_2_), 2.72 (1H, dd, *J* = 18.2, 6.6 Hz CH_b_, (succinic anhydride CH_2_)), 1.84–1.75 (1H, m, CH_a_ (CH_2_CH)), 1.67–1.56 (1H, m, CH_b_ (CH_2_CH)), 1.45–1.34 (2H, m, CH_2_ (CH_2_CH_2_CH_2_)), 1.15 (9H, t, *J* = 7.0 Hz, 3CH_3_ (OEt)), 0.63–0.50 (2H, m, CH_2_ (SiCH_2_)).

**PEG**. ^1^H NMR (DMSO-d_6_, 400 MHz) *δ* 12.4–11.8 (1H, br. s, COOH), 8.0–7.5 (2H, br. s, NH_2_), 3.68 (2H, t, *J* = 5.1 Hz, CH_2_ (H_2_NCH_2_)), 3.63–3.45 (m, (CH_2_CH_2_O)_n_ and H_2_NCH_2_CH_2_), 2.98 (2H, t, *J* = 5.2 Hz, CH_2_CH_2_COOH), 2.44 (2H, t, *J* = 6.5 Hz, CH_2_CH_2_COOH).

**Conjugate 5**. ^1^H NMR (DMSO-d_6_, 500 MHz) *δ* 12.7–11.6 (2H, br. s, COOH), 7.80 (1H, s, NHCO), 6.5–5.2 (3H, br. s, Si(OH)_3_)), 3.65 (2H, t, *J* = 4.8 Hz, CH_2_ (H_2_NCH_2_CH_2_)), 3.62–3.47 (m, (CH_2_CH_2_O)_n_ and H_2_NCH_2_CH_2_), 3.44 (6H, q, *J* = 7.0 Hz, 3CH_2_ (EtOH)), 3.23–3.15 (1H, m, CH), 3.03 (1H, dd, *J* = 18.2, 9.7 Hz, CH_a_ (CH_2_ succinic acid)), 2.97 (2H, dt, *J* = 5.5, 5.4 Hz, CH_2_CH_2_COOH), 2.71 (1H, dd, *J* = 18.3, 6.5 Hz, CH_b_ (CH_2_ succinic acid)), 2.44 (2H, t, *J* = 6.4 Hz, CH_2_CH_2_COOH), 1.80–1.72 (1H, m, CH_a_ (CH_2_-CH)), 1.66–1.56 (1H, m, CH_b_ (CH_2_CH)), 1.39 (2H, m, CH_2_CH_2_CH_2_), 1.06 (9H, t, *J* = 7.0 Hz, 3CH_3_ (EtOH)), 0.49–0.31 (2H, m, SiCH_2_).

### 3.5. Synthesis of MNPs@PEG (MNPs ***6**–**8***)

TESPSA (19 µL, 67 µmol) was added to a solution of PEG (20.1 mg, 6.7 µmol) in dry DMSO (1.11 mL); the reaction mixture was stirred for 1 h at 25 °C. The resulting solution (1000, 100, or 10 µL) was added to a colloidal solution of MNPs **1** (20 mg) in 70% EtOH (7 mL) at 40 °C. The reaction mixture was sonicated in an ultrasound bath and stirred with an overhead stirrer for 4 h at 40 °C and 16 h at 20 °C, then centrifuged (15,000 rpm) for 10 min. The resulting MNPs were separated from supernatant, washed with water (3 × 10 mL), and resuspended in water (5 mL) to afford colloidal solutions of MNPs **6**–**8**.

### 3.6. Synthesis of MNPs@PEG-Dox (MNPs ***9**–**11***)

A solution of Dox × HCl (0.95 mg) in water (200 µL) was added to a suspension of MNPs **6–8** (0.95 mg) in water (1 mL). The reaction mixture was sonicated in an ultrasound bath, kept for 20 h at room temperature, and centrifuged at 15,000 rpm for 10 min. The resulting MNPs were separated and washed with water (3 × 1 mL), and resuspended in water (1 mL) to afford MNPs **9**–**11**.

The amount of Dox in supernatant was determined by UV spectrometry (λ_max_ = 490 nm) by analogy with Ref. [33]. The Dox loading efficiency (LE, wt.%) and loading content (LC, wt.%) were calculated by Formulas (1) and (2), respectively:LE = (*m*_Dox load_ − *m*_Dox_) × 100%/*m*_Dox load_,(1)
LC = (*m*_Dox load_ − *m*_Dox_) × 100%/*m*_nanocomposite_,(2)
where *m*_Dox load_ is the mass (mg) of Dox loaded in the reaction, *m*_Dox_ is the amount (mg) of Dox in supernatant, *m*_nanocomposite_ is the mass (mg) of Dox-containing nanocomposite.

### 3.7. Characterization of Nanocomposites

The ^1^H NMR spectra were recorded on a Bruker Avance 500 (500 MHz) or Bruker DRX-400 (400 MHz) instruments in DMSO-d_6_ with TMS as an internal reference at ambient temperature. The IR spectra were recorded on a Perkin Elmer Spectrum Two FT-IR spectrometer (Perkin Elmer, Waltham, MA, USA) equipped with the ultra-attenuated total reflection (UATR) (MNPs **1–5**) on the diamond crystal, and the diffuse reflectance infrared Fourier transform spectroscopy (DRIFT) (MNPs **6–8**) accessories. Microanalyses were performed using a ЕuroEA 3000 automatic analyzer (EuroVector Instruments & Software, Milan, Italy). The EDX spectra and Fe and Si fractions were determined on an EDX-7000 X-ray fluorescence spectrometer (Shimadzu, Kyoto, Japan). The amount of Dox in aqueous solutions was measured using a UV 2600 spectrometer (Shimadzu, Kyoto, Japan). DLS characterization of aqueous solutions was carried out on a Malvern Zetasizer Nano ZS instrument (Malvern Instruments, Malvern, UK). The magnetic properties were studied on a magnetic vibromagnetometer with fields up to 25 kOe at room temperature. Transmission electron microscopy (TEM) images were obtained on a Tecnai G2 30 Twin transmission electron microscope (Thermo Fisher Scientific, Waltham, MA, USA).

### 3.8. Determination of Specific Absorption Rate (SAR) and Intrinsic Loss Power (ILP) of MNPs ***6*** and ***9***

The SAR of MNPs in an aqueous solution was measured using the TOR UltraHT facility to study the characteristics of local magnetic hyperthermia (Nanomaterials, Tambov, Russia). This instrument allows experiments to be carried out at a single magnetic field frequency of 230 kHz. For the experiments, we used colloidal solutions of MNPs **6** and **9**; Fe concentration was determined using ferrozine method [4,40]. The measurements were carried out for colloidal solutions of MNPs with a [Fe] concentration of 10 mg/mL in a total volume of 500 µL at a magnetic field frequency of 230 kHz and a magnetic induction of 0.27 kOe. Temperature was measured using fiber optic temperature sensor (Fotemp-H, Optocon, Dresden, Germany). 

As the first linear trend (0–40 s) is hindered by the inertia of the convective heat transfer in the water volume, it seems that the best temperature rate estimator is in the 40–80 s time interval. The calculation of SAR (W/g) for the experimental sample was carried out according to Formula (3):SAR = d*T*/d*t*·*m*_1_/*m*_Fe_·*C*,(3)
where d*T*/d*t* is the sample heating rate for the 40–80 s time interval, which was determined by the slope of the initial section of the suspension heating curve after AMF was switched on, K/s; *m*_l_ is the suspension weight, g; *m*_Fe_ is the mass of nanoparticles in suspension (in terms of Fe concentration), g; *С* is the specific heat capacity of the suspension, J/g·K. Taking into account that the contribution of nanoparticles to the specific heat capacity of the suspension can be neglected, this value is assumed to be equal to the specific heat capacity of water and is 4.18 J/g·K.

The calculation of the intrinsic loss power (ILP, nH m^2^/kg) was carried out according to Formula (4):ILP = SAR/(*H*^2^ *f*),(4)
which is the SAR value normalized to the frequency and the AMF amplitude.

### 3.9. Study of Dox Desorption from MNPs ***9–11*** during AMF Treatment of Their Aqueous Colloidal Solutions

In order to study the desorption of Dox from the surface of MNPs **9** induced by AMF, the samples were resuspended in sodium phosphate buffer with pH 5.8 or 7.4 and sonicated for 15 s. Then, 300 µL of the suspension with a [Fe] concentration of 1 mg/mL was added to 0.6 mL test tubes and placed into a TOR Ultra HT device. AFM exposure was performed at a frequency of 230 kHz (0.27 kOe); the temperature in the device chamber was maintained at 30 °C. Control samples were placed in a thermostat and incubated at 30 °C. The free release samples were incubated on a rotator (Biosan) at 28 rpm, 37 °C for 24 h. After the completion of the incubation, the particles were sedimented by centrifugation, and the fluorescence intensity of the supernatant was assessed at λ_ex_ = 480 nm and λ_em_ = 590 nm using an Infinite 200 PRO multimode plate reader (Tecan, Grodig, Austria). In order to quantify the released Dox, calibration plots of Dox fluorescence intensity in sodium phosphate buffers (pH = 5.8 and 7.4) versus concentration were preliminarily plotted (Figure 4).

### 3.10. Assessment of Cytotoxicity of MNPs ***6*** and MNPs ***9***

The study of the cytotoxic effect of MNPs **6** and MNPs **9** was carried out using the following tumor cell lines: human breast adenocarcinoma MDA-MB231, human hepatocellular carcinoma HepG2, murine colorectal carcinoma CT-26, murine melanoma B16, and murine mammary carcinoma 4T1 by the MTT (3-(4,5-dimethylthiazol-2-yl)-2,5-diphenyltetrazolium bromide) assay [4,41]. Cell lines MDA-MB231, HepG2, 4T1, and B16 were cultured in complete DMEM/F12 medium (Gibco, Grand Island, NY, USA) supplemented with 10% FBS (Gibco, Paisley, UK), 1 × PenStrep (Gibco, Grand Island, NY, USA), and 1 × Glutamax (Gibco, Grand Island, NY, USA); CT26 cells, in complete RPMI 1640 medium (Gibco, Grand Island, NY, USA) supplemented with 10% FBS, 1 × PenStrep, and 1 × Glutamax, 1 mM HEPES (7.4) in CO_2_ incubator at 37 °C. TrypLE (Gibco, Grand Island, NY, USA) was used for dissociating adherent cells from the surface of plastic flasks.

For the MTT assay, 4 × 10^3^ cells (CT26, 4T1, and B16) or 8 × 10^3^ cells (MDA-MB231 and HepG2) in 90 µL of complete medium were plated into the wells of a 96-well plate. After 24 h, 10 µL of a suspension of nanoparticles at appropriate concentrations was added to the cells; doxorubicin was used as a positive control; sterile water as a negative control; 0.1% Triton X-100, as a nonspecific positive control. The plates were incubated for 24 and 48 h in a CO_2_ incubator. Then, the medium was removed from the wells, and 100 µL of a complete medium with the MTT reagent (PanEco, Moscow, Russia) at a concentration of 0.5 mg/mL was added and incubated for 2 h at 37 °C. Formazan crystals were dissolved in DMSO (200 µL) and the absorbance was measured in the wells at 540 nm and a reference wavelength of 620 nm (Infinite 200 PRO, Tecan, Groding, Austria).

The calculation of cell viability (%) was performed relative to cell viability in the control according to the Formula (5):Cell viability (%) = (*A*_ex_/*A*_c_) × 100,(5)
where *A*_ex_ is the optical density in wells with test substances, *A*_c_ is the optical density of negative control wells.

The IC_50_ values were determined by nonlinear regression analysis using GraphPad Prism7.

### 3.11. Assessment of Cytotoxicity of MNPs ***6*** and ***9*** under Application of AMF

In order to assess the effect of AMF on the cytotoxicity of MNPs, 4 × 10^3^ 4T1 cells were seeded into 8-well strips (previously coated with poly-L-lysine, Sigma) in 90 µL of a complete DMEM/F12 medium. After 24 h, 10 µL of the nanoparticle suspension to a final concentration of 5 µg [Fe]/mL (*n* = 4) was added to the cells; water was added to the control wells (*n* = 4). The strips were placed in a TOR Ultra HT device and incubated for 1 h at 230 kHz, 0.27 kOe, 30 °C. Control strips (without AMF exposure) were incubated in a thermostat at 30 °C for 1 h. After incubation, strips were placed in a CO_2_ incubator. After 24 h, cell viability was assessed by the MTT assay.

### 3.12. Statistical Analysis

Statistical data processing was carried out using GraphPad Prism7 (GraphPad Software, San Diego, CA, USA). Data were presented as Mean ± SD. One-way ANOVA followed by Dunnett’s multiple comparison test was used to determine the significance of differences between several groups. Differences at *p* < 0.05 were considered statistically significant.

## 4. Conclusions

Thus, we have developed novel nanocomposite magnetic-responsive materials based on Fe_3_O_4_ and SiO_2_/PEG with a high level of doxorubicin loading. Their cytotoxicity towards various cancer cells (MDA-MB231, HepG2, 4T1, CT26, and B16) was shown. The therapeutic effect of Dox released from the MNPs surface due to local magnetic hyperthermia was demonstrated in in vitro experiments. The application of an alternating magnetic field of 0.27 kOe with a frequency of 230 kHz, close to the safety limit of 6.25 × 10^7^ Oe Hz, leads to a 2.5-fold increase in the cytotoxic effect of the material against 4T1 tumor cells within 1 h. We believe that the obtained materials can be used to design magnetic-responsive materials with controlled drug release for the targeted synergistic magnetic hyperthermia/cancer chemotherapy.

## Data Availability

Not applicable.

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
