# Peer review of "Magnetic-Responsive Doxorubicin-Containing Materials Based on Fe3O4 Nanoparticles with a SiO2/PEG Shell and Study of Their Effects on Cancer Cell Lines"

_ijms, 2022, doi:10.3390/ijms23169093_

Round 1
Reviewer 1 Report
I find the paper correctly written, but not ambitious.
The problem I see is that it does not explore the magnetic responsiveness of the synthesized material in much more details. Was the field frequency of 230 kHz the only one obtainable by the TOR UltraHT facility? Putting differently, the Dox desorption from MNP 9-11 and cell viability should have been studied as functions of magnetic field parameters and the corresponding conclusions drawn on the mechanism of the Dox release under the alternating magnetic field.
In addition, this is yet another paper on the synthesis of "potentially useful material for targeted cancer chemotherapy".
Author Response
Dear Reviewer 1,
The authors are very grateful to you for valuable remarks and comments. We believe that our responses to these comments and revisions we made will improve our manuscript for publication in the International Journal of Molecular Sciences.
Reviewer 1. Comments and Suggestions for Authors:
- The problem I see is that it does not explore the magnetic responsiveness of the synthesized material in much more details. Was the field frequency of 230 kHz the only one obtainable by the TOR UltraHT facility? Putting differently, the Dox desorption from MNP 9-11 and cell viability should have been studied as functions of magnetic field parameters and the corresponding conclusions drawn on the mechanism of the Dox release under the alternating magnetic field.
Reply: The TOR UltraHT facility allows operating only at the specified frequency, therefore, unfortunately, at the moment we are not able to perform studies on desorption and cytotoxic effects at other field parameters. We have studied the effect of three magnetic field modes on the Dox release from the nanomaterial and found the conditions that provide the best Dox desorption.
In the text of the manuscript, subsection 2.2. Dox Release under Exposure to an Alternating Magnetic Field (page 6, lines 179-181), we added the following sentence: “Based on this, it can be concluded that conditions (continuous exposure to an alternating magnetic field for 1 h), which ensure the constancy of the particle temperature at the maximum level, make it possible to achieve the best results of Dox desorption.”
- In addition, this is yet another paper on the synthesis of "potentially useful material for targeted cancer chemotherapy".
Reply: The authors agree with the Reviewer’s comment that this is another paper on the synthesis of “potentially useful material for targeted cancer chemotherapy”. However, due to the fact that the next stage of research, namely, experiments in vivo, is very complex and should take into account a large number of factors (administration regimens, drug doses, magnetic field parameters, time intervals providing the best effect, etc.), at this stage it seems important to develop the best composition of the nanocomposite material.
On behalf of all coauthors.
Sincerely yours,
Dr. Alexander M. Demin

Reviewer 2 Report
Dear authors,
I recommend the paper for publication provided the following issues are addressed:
1. Line 277: par. 3.8 title"...MNPs 6" seem incomplete
2. Line 283: the magnetic field intensity amplitude value of 27kOe is unrealistic at 230kHz. Please check.
3. Please add to the Electronic Supplementary Info: a) the MNPs diameter histogram as obtained from TEM, and b) the proof for magnetite from electron difractograms.
4. Just a suggestion: You underestimate the MNPs SAR. When measuring MNPs SAR in water, the first linear trend (0-40s) is hindered by the inertia of the convective heat transfer in the water volume. Therefore, a better temperature rate estimator is in the 40-80s time interval.
Best regards,
Author Response
Dear Reviewer 2,
The authors are very grateful to you for valuable remarks and comments. We believe that our responses to these comments and revisions we made will improve our manuscript for publication in the International Journal of Molecular Sciences.
Reviewer 2. Comments and Suggestions for Authors:
- Line 277: par. 3.8 title "...MNPs 6" seem incomplete.
Reply: We revised the title of subsection 3.8, see page 9, lines 281-282.
- Line 283: the magnetic field intensity amplitude value of 27 kOe is unrealistic at 230 kHz. Please check.
Reply: The authors thank the Reviewer 2 for noticing the typo we made. The correct value is 0.27 kOe. The text of the manuscript has been appropriately corrected (lines 289, 311, and 355).
- Please add to the Electronic Supplementary Info: a) the MNPs diameter histogram as obtained from TEM, and b) the proof for magnetite from electron diffractograms.
Reply: In the Supplementary Materials, we have added relevant information (Figures S1 and S2, Table S3). References to materials added to the Electronic Supplementary are inserted into the text of the manuscript (see lines 156-158).
- Just a suggestion: You underestimate the MNPs SAR. When measuring MNPs SAR in water, the first linear trend (0-40 s) is hindered by the inertia of the convective heat transfer in the water volume. Therefore, a better temperature rate estimator is in the 40-80 s time interval.
Reply: The authors thank the Reviewer 2 for the suggestion. We recalculated the value of SAR and added the obtained data in Figure 3b. See also the revised text in subsection 3.8, lines 291-293, 295.
On behalf of all coauthors.
Sincerely yours,
Dr. Alexander M. Demin

Round 2
Reviewer 1 Report
Although major revisions were asked for, only a minor revision has been done.
Author Response
Dear Reviewer 1,
Following your recommendations, we have made serious revisions (highlighted in yellow) to the manuscript. Previous amendments remain highlighted in green.
We have revised the description of the magnetic responsiveness of the synthesized material. Based on previously obtained results, we have described in more detail the mechanism of the Dox release under an alternating magnetic field. In this regard, the list of the cited references has been expanded. We have improved the research design: for example, we have made significant changes to sections 2.1, 2.2, and 2.3; data describing the cytotoxicity of synthesized MNPs and the effect of AMF in section 3. Materials and Methods, we divided into two subsections (3.10 and 3.11) for a clearer presentation of the results. The conclusions were also reformulated.
The authors agree with the Reviewer’s comment that “the Dox desorption from synthesized MNPs and cell viability should have been studied as functions of magnetic field parameters”; however, the TOR UltraHT facility we use allows experiments to be performed at a single magnetic field frequency of 230 kHz. An appropriate clarification has been made in subsection 3.8. A more detailed study of Dox desorption from MNPs and its effect on cell viability depending on magnetic field parameters is planned to be carried out in the further experiments.
On behalf of all coauthors.
Sincerely yours,
Dr. Alexander M. Demin
